# Comparison of Osteoconductive Ability of Two Types of Cholesterol-Bearing Pullulan (CHP) Nanogel-Hydrogels Impregnated with BMP-2 and RANKL-Binding Peptide: Bone Histomorphometric Study in a Murine Calvarial Defect Model

**DOI:** 10.3390/ijms24119751

**Published:** 2023-06-05

**Authors:** Cangyou Xie, Fatma Rashed, Yosuke Sasaki, Masud Khan, Jia Qi, Yuri Kubo, Yoshiro Matsumoto, Shinichi Sawada, Yoshihiro Sasaki, Takashi Ono, Tohru Ikeda, Kazunari Akiyoshi, Kazuhiro Aoki

**Affiliations:** 1Department of Basic Oral Health Engineering, Graduate School of Medical and Dental Sciences, Tokyo Medical and Dental University, Tokyo 113-8549, Japan; pomelo0110@gmail.com (C.X.); fatma.rashed@hotmail.com (F.R.); yosuke.sasaki@envistaco.com (Y.S.); masud_khan.bhoe@tmd.ac.jp (M.K.); drqijia@outlook.com (J.Q.); 2Department of Oral Pathology, Graduate School of Medical and Dental Sciences, Tokyo Medical and Dental University, Tokyo 113-8549, Japan; tohrupth.mpa@tmd.ac.jp; 3Department of Oral Biology, Faculty of Dentistry, Damanhour University, Damanhour 22511, Egypt; 4Department of Orthodontic Science, Graduate School of Medical and Dental Sciences, Tokyo Medical and Dental University, Tokyo 113-8549, Japan; y.matsumoto.orts@tmd.ac.jp (Y.M.); t.ono.orts@tmd.ac.jp (T.O.); 5Department of AI Technology Development, Graduate School of Medical and Dental Sciences, Tokyo Medical and Dental University, Tokyo 113-8549, Japan; yshidsc@tmd.ac.jp; 6Department of Polymer Chemistry, Graduate School of Engineering, Kyoto University, Kyotodaigaku Katsura, Kyoto 615-8510, Japan; sawada@bio.polym.kyoto-u.ac.jp (S.S.); sasaki.yoshihiro.8s@kyoto-u.ac.jp (Y.S.); akiyoshi.kazunari.2e@kyoto-u.ac.jp (K.A.)

**Keywords:** bone regeneration, scaffold material, bone morphogenetic protein (BMP)-2, receptor activator of NF-κB ligand (RANKL)-binding peptide, cholesterol-bearing pullulan (CHP) nanogels

## Abstract

The receptor activator of NF-κB ligand (RANKL)-binding peptide is known to accelerate bone morphogenetic protein (BMP)-2-induced bone formation. Cholesterol-bearing pullulan (CHP)-OA nanogel-crosslinked PEG gel (CHP-OA nanogel-hydrogel) was shown to release the RANKL-binding peptide sustainably; however, an appropriate scaffold for peptide-accelerated bone formation is not determined yet. This study compares the osteoconductivity of CHP-OA hydrogel and another CHP nanogel, CHP-A nanogel-crosslinked PEG gel (CHP-A nanogel–hydrogel), in the bone formation induced by BMP-2 and the peptide. A calvarial defect model was performed in 5-week-old male mice, and scaffolds were placed in the defect. In vivo μCT was performed every week. Radiological and histological analyses after 4 weeks of scaffold placement revealed that the calcified bone area and the bone formation activity at the defect site in the CHP-OA hydrogel were significantly lower than those in the CHP-A hydrogel when the scaffolds were impregnated with both BMP-2 and the RANKL-binding peptide. The amount of induced bone was similar in both CHP-A and CHP-OA hydrogels when impregnated with BMP-2 alone. In conclusion, CHP-A hydrogel could be an appropriate scaffold compared to the CHP-OA hydrogel when the local bone formation was induced by the combination of RANKL-binding peptide and BMP-2, but not by BMP-2 alone.

## 1. Introduction

Local bone regeneration is essential for maintaining healthy bone structure after trauma, or surgical bone defect [1,2]. External supporting elements are needed for bridging and facilitating bone regeneration in different clinical conditions, such as periodontal regeneration, cleft palate, and pathological conditions that prevent normal bone healing [3,4]. Scaffolds are one of the essential elements that are used for tissue repair and bone regeneration [5,6]. They boost cell–cell communication, growth factors and/or signaling molecules exchange. Scaffolds can come in different shapes, materials, structures, and pore sizes, which can affect the quality and quantity of newly formed bone [7,8,9,10,11]. So far, the fabrication of bone scaffolds with the desirable chemical properties for bone tissue engineering is still challenging. Many kinds of bone scaffolds for bone tissue regeneration are developed; collagen [12] and ceramic type scaffolds, including tricalcium phosphate (TCP) [13] and hydroxyapatite (HA) [14], and polymer-type scaffolds, including poly lactic-co-glycolic acid (PLGA) [15] and polycaprolactone (PCL) [16] have already been used in the clinical application. Chemical surface functionalization of the scaffold is one of the approaches to improve its performance for biological molecules, such as drugs, growth factors, and peptides [17]. Researchers in the field of interface science have developed the modifications of scaffolds surface for peptide-induced bone tissue regeneration [17,18,19]. In a previous study, polydopamine (pDA)-coated PCL nanofiber scaffold was shown to enhance osteoconductivity, followed by covalent immobilization of bone morphogenetic protein (BMP)-7-derived peptides onto the pDA-coated nanofiber scaffold surface [18]. Another later study has shown that decellularized extracellular matrix (ECM)-modified true bone ceramic (TBC) scaffold induced bone repair when aspartic acid-modified BMP-2 peptide was immobilized onto the surface-modified TBC scaffold [19]. The development of scaffold materials can enhance the efficiency of the attachment of the signaling molecule and improve local bone regeneration [20]; however, it is well known that peptides tend to aggregate with an inactive form, leading to the suppression of their bioactivity [21]. Hence, it is necessary to maintain the three-dimensional (3D) structure of the functional peptide for exerting its bioactivity when we use peptides as a bone anabolic reagent.

Hydrogels are promising candidates amongst bone scaffolds and have been conducted in the clinical trial phases [22]. A cholesterol-bearing pullulan (CHP) nanogel-hydrogel, one of the hydrogels, has shown the ability to inhibit the aggregation of a functional peptide [23]. This CHP nanogel-based hydrogel can arrange and maintain the 3D structure of the peptide in a functional form, thereby exhibiting its original bioactivity [24]. Since the CHP nanogel is shown to have a chaperone-like function, which prevents aggregation of the encapsulated peptide as mentioned here, the CHP nanogel-hydrogel could be an expected scaffold for peptide-induced bone regeneration.

The hydrogel containing CHP-A nanogel, one of the reactive CHPs, was shown to be a potential scaffold for tongue muscle regeneration [25]. Recently, the research is focusing on using two signaling molecules that offers the synergistic effects and different release rates from the scaffold [26]. In our local bone regeneration study, changing the physical structure of the CHP-A hydrogel was shown to be able to induce local bone formation, especially when it was impregnated with a combination of two signaling molecules: BMP-2 and the receptor activator of NF-κB ligand (RANKL)-binding peptide (OP3-4) [27]. Using this combination of two-signaling-molecules, synergistic effects on bone formation have been already proven in vitro and in vivo [28].

Another type of reactive CHP, the hydrogel containing CHP-OA nanogel, has been developed and was proven to be a good scaffold for tissue regeneration [29]. The CHP-OA nanogel has an extra urethane bond linked to the ester bond in the CHP-A nanogel. In our previous study, we have already shown that the CHP-OA nanogel-hydrogel can offer the effect of a sustained peptide release to prevent bone resorption in a low calcium-diet-induced osteopenic model [30]. In addition, an osteoclastogenesis assay was performed to clarify a sustained release of the RANKL-binding peptide from the CHP-OA nanogel, compared to conventional CHP nanogels, which were not chemically crosslinked with the 4-armed polyethylene glycol thiol (PEG-SH) [30]. Although the inhibitory effects of bone resorption of the CHP-OA nanogel-hydrogel were shown in vitro and in vivo, it is still unknown whether the CHP-OA nanogel-hydrogel, as a scaffold material, can offer osteoconductive ability when incorporated with BMP-2 and the RANKL-binding peptide.

Therefore, this study aimed to compare the osteoconductive ability of two types of CHP nanogel-hydrogels: CHP-A and CHP-OA. These nanogel-hydrogels were considered as scaffold materials in combination with BMP-2 and the RANKL-binding peptide, OP3-4 in a murine calvarial defect model.

## 2. Results

### 2.1. Chemical Structure of Two Types of CHP Nanogels

CHP nanogel was modified to form two reactive CHP nanogels. The CHP-A nanogel is linked to the hydroxyl group of pullulan by an ester bond, and the CHP-OA nanogel is linked to the pullulan chain by an urethane linker, which functioned as a hardly hydrolyzed material in a stable status (Figure 1). The degradation rate of the CHP-OA nanogel-crosslinked PEG hydrogel (CHP-OA nanogel-hydrogel) over time in buffer solutions with three different pH levels (6.0, 7.4, and 8.0) was examined (Appendix A). The fastest degradation rate was seen in the alkaline buffer (pH 8.0). The CHP-OA nanogel-hydrogel began degrading slowly and reached 100% on day 22. In the neutral pH 7.4 solution, it degraded slower than in the alkaline buffer and began degrading around day 30, reaching 100% on day 55. The degradation rate barely changed at pH 6.0, even though 65 days had passed. On the other hand, previously published data of the CHP-A nanogel-crosslinked PEG hydrogel (CHP-A nanogel-hydrogel) began degrading from day 1, reaching 70% before day 10 in the neutral solution (Appendix A) [27].

### 2.2. Radiological Analysis of Calcified Tissue in the Calvarial Bone Defect In Vivo

The 3D reconstruction images of μCT on week 4 after the operation showed that no newly calcified tissue was observed when CHP-A scaffold and CHP-OA scaffold groups were used alone for bone regeneration (Figure 2A); however, when CHP-A and CHP-OA scaffold groups were impregnated with BMP-2, newly calcified tissue was visually observed on week 4 after the operation (Figure 2A). Quantitative analysis of the calcified tissue at week 4 after scaffold placement was significantly higher when both scaffolds were impregnated with BMP-2, compared to those without BMP-2 (Figure 2B). To examine the newly calcified tissue shown in the μCT images, peripheral quantitative computed tomography (pQCT) was performed to analyze the bone mineral content and bone mineral density at the bone defect site. pQCT analyses showed that the total bone mineral content and bone mineral density of both scaffolds were significantly higher when impregnated with BMP-2 compared to those without BMP-2 (Figure 2C,D).

We then examined the osteoconductive ability of CHP-A and CHP-OA nanogel-hydrogel scaffolds when impregnated with BMP-2 alone, BMP-2+Vehicle, or BMP-2+OP3-4. We observed the changes in the newly calcified tissue, using 3D reconstruction images of μCT in the calvarial defect during the time period of 4 weeks after the operation. Newly calcified tissue was observed from the second week and continued forming to week 4 in almost all groups; however, less newly calcified tissue was observed in the CHP-OA group compared to the CHP-A group (Figure 3A). Micro-CT images showed that the calcified tissue in the calvarial defect was observed only at the periphery of the defect when BMP-2 alone or BMP-2+Vehicle was impregnated in both scaffolds on week 4 (Figure 3A). When BMP-2 and OP3-4 were impregnated in both scaffolds, the CHP-OA group showed less calcified tissue than the CHP-A group, which almost fully covered the bone defect (Figure 3A). Through quantitative analysis, newly calcified tissue showed no significant difference between CHP-A and CHP-OA scaffold groups when BMP-2 was used without OP3-4 peptide. Thereafter, soft X-ray analyses were performed to confirm the calcified tissue shown in the μCT images. The CHP-OA group revealed less radio-opaque area than the CHP-A group when BMP-2 and OP3-4 were used (Figure 4).

### 2.3. Bone Histomorphometric Analyses at the Calvarial Bone Defect

Von Kossa staining was performed to show the newly calcified bone, using undecalcified sections at the bone defect site. The calcified bone area (black stained area) was similar in CHP-A and CHP-OA scaffold groups when both scaffolds were impregnated with BMP-2 alone or BMP-2+Vehicle, as shown in Figure 5A. However, the calcified bone area was visually different when BMP-2 and OP3-4 were impregnated in both scaffolds (Figure 5A). The calcified bone area was then measured and quantified (Figure 5A). The newly calcified bone area and average bone thickness showed no significant difference when BMP-2 was used without OP3-4 in both scaffolds (Figure 5B,C), but when OP3-4 was used, the CHP-OA group showed significantly less newly calcified bone area and average bone thickness than the CHP-A group (Figure 5B,C).

Calcein and alizarin injections were performed on day 10 and 18, and the fluorescence-labeled area, which represents bone formation activity, was measured using undecalcified sections at the bone defect site. The areas of calcification during calcein administration were green, and those during alizarin administration were red (Figure 6A). Mostly green calcein labeling was observed in the CHP-A group, yet mostly red alizarin labeling was observed in the CHP-OA group. No apparent difference in the fluorescence-labeled area was observed when CHP-A and CHP-OA groups were impregnated with BMP-2 alone or BMP-2+Vehicle, but there was a significant difference only when BMP-2 and OP3-4 were impregnated in both scaffolds (Figure 6A). We then measured the fluorescence-labeled area in the white rectangle (Figure 6A), using the sum of two fluorescent labeling. Similar results were shown when BMP-2 was used without OP3-4 in both scaffolds, but the CHP-OA group obtained significantly lower bone formation activity in comparison to the CHP-A group when BMP-2 and OP3-4 were used (Figure 6B).

## 3. Discussion

We compared two scaffold materials, the hydrogels crosslinked by the CHP-A nanogel or the CHP-OA nanogel impregnated with either or a combination of BMP-2 and RANKL-binding peptide (OP3-4), to clarify their osteoconductive ability on local bone regeneration. The degradation rate of the CHP-OA nanogel-hydrogel was analyzed (Appendix A) and compared to the degradation rate of the CHP-A nanogel-hydrogel results shown in our previous study [27] (Appendix A). In the neutral buffer solution, the CHP-A nanogel-hydrogel started to degrade around day 3, and nearly 70% of the scaffold was degraded around day 10 (data shown in Xie et al. 2022 [27]), while the CHP-OA nanogel-hydrogel began to degrade gradually around day 28. Our degradation rate analyses revealed that the CHP-OA nanogel-hydrogel has a slower rate in comparison to the CHP-A nanogel-hydrogel (Appendix A). This can be attributed to the presence of an urethane bond in the CHP-OA chemical structure (Figure 1A).

Through radiological analyses, no newly formed bone was observed when we used the CHP-A scaffold or the CHP-OA scaffold alone to induce bone formation (Figure 2A). This proved that the scaffold itself had no osteoconductive effect. The amount of bone regenerated in the CHP-OA scaffold was significantly less than that in the CHP-A scaffold when the RANKL-binding peptide was used (Figure 3B). Since the effect of the RANKL-binding peptide was shown to be most effective at the early stage of osteoblast differentiation [28], the appropriate timing of the scaffold degradation is essential. As we have shown that the CHP-OA has a much slower degradation rate compared to the CHP-A in our degradation rate analyses (Appendix A), this very slow degradation rate of the CHP-OA might hinder the release of the RANKL-binding peptide at the appropriate timing and consequently lower its bone regenerative capacity.

There was no significant difference in the amount of bone regenerated between the CHP-A and the CHP-OA scaffolds when both scaffolds were impregnated with BMP-2 alone, but when the scaffolds were impregnated with both the BMP-2 and OP3-4 peptide, bone formation regenerated in the CHP-OA scaffold was much less compared to the CHP-A scaffold (Figure 2 and Figure 3). We might explain this different response on osteoconductivity to the relationship between the amount of releasing molecules and the binding affinity of the signaling molecules to the scaffolds. The hydrogels possess a porous micro-sized network and an adjustable degradation rate [31]. So far, it has been shown that the release profile of proteins from scaffolds can be affected by degradation [32,33], stimuli [34,35], covalent bonds [36,37], and affinity with binding sites [38,39,40]. In our present study, affinity with binding sites could be an important factor for regulating the release profiles of impregnated molecules. The binding affinity of the OP3-4 peptide with the scaffold could be weaker than that of BMP-2 since the RANKL-binding peptide, OP3-4, whose molecular weight (MW) was approximately 1.4 kDa smaller than that of BMP-2 (MW: 26 kDa), could have fewer binding sites to the nanogel network in the scaffolds than BMP-2. Because the CHP nanogel-hydrogel itself has a nano-sized network [24], larger BMP-2 protein was trapped in the nano-sized network. The smaller molecule OP3-4 might be able to move around easily in the nano-sized network. Actually, our previous study showed that the RANKL-binding peptide, OP3-4, could not be detected by using high-speed atomic force microscopy, while the soluble RANKL molecule (MW: 20 kDa), which was similar molecular size to the BMP-2 molecule, could be detected, and the size of the RANKL molecule was approximately 5 nm [28]. Therefore, the release profile of the BMP-2 protein from the scaffold could not be affected by the difference in the degradation rate of scaffolds, while that of the OP3-4 peptide from the scaffold could be influenced by the degree of the degradation rate. Since the degradation rate was much slower in the CHP-OA scaffold compared to the CHP-A scaffold, the OP3-4 peptide could release from the CHP-OA scaffold later than from the CHP-A scaffold, leading to the difference in regenerated bone amount. Consequently, the difference in the degradation rate could directly affect the release of the peptide, causing the difference in the peptide-enhanced bone formation, while the release of BMP-2 could not be influenced by the degradation rate. Taken together, our results revealed for the first time that a release mechanism from the CHP nanogel-hydrogel scaffold could be a “size-dependent release”, affecting local bone regeneration.

We examined the effects of an alkaline buffer, the vehicle of the RANKL-binding peptide, because the pH level of examined buffer is known to affect the degradation rate and consequently controls the release of peptides from the hydrogel scaffolds [41]; however, the high pH vehicle did not affect the amount of bone formation significantly in both CHP-A and CHP-OA cases, suggesting that the vehicle did not affect the degradation rate and the release profile of these two CHP nanogel materials (Figure 3 and Figure 4).

To acquire a quantitative value of µCT, one of the critical factors is threshold setting [42], which can separate hard tissue and soft tissue. Researchers can adjust the threshold value, which results in different bone reconstruction images of µCT [42]. This indicates that the regenerated bone of the defect may sometimes not be a real calcified tissue. Thus, we performed a soft X-ray investigation, and confirmed that the radio-opaque area of BMP-2 and OP3-4 with the CHP-A covered most of the defect (Figure 4), as shown in the µCT images (Figure 3A). In addition, we performed the von Kossa staining [43], which is a method to illustrate mineralization that appears in the undecalcified sections to clarify whether the regenerated bone of defect is real calcified tissue or not. The von Kossa staining in the CHP-OA scaffold showed significantly less osteoconductive ability than in the CHP-A scaffold only when BMP-2 and OP3-4 were impregnated in both scaffolds (Figure 5B).

Here, we used “labeled bone area” to show the bone formation activity. To obtain information on bone formation activity, bone fluorescent labeling was performed with calcein and alizarin injections (Figure 6). The bone formation rate, which shows the calcification-ability of osteoblasts, is a standard parameter calculated by using the distance data between two parallel fluorescent label lines to determine how much bone has formed during the interval between the administrations of the different fluorescent injections [44]; however, bone regeneration in the calvarial bone defect site was fast, producing a woven bone, thereby resulting in broad fluorescent lines (Figure 6A). This caused difficulty in measuring the distance between the two parallel label lines; therefore, we measured the labeled bone area instead of using bone formation rate, the standard parameter of bone histomorphometry (Figure 6). By analyzing bone formation activity, the CHP-OA obtained significantly lower bone formation activity compared to the CHP-A only when BMP-2 and OP3-4 were impregnated (Figure 6B).

Our results revealed that the CHP nanogel-hydrogels could make the unstable peptide work as a bone anabolic reagent even under the limitation of bone scaffolds and lack of physicochemical properties [45,46]. Since the physiochemical properties are shown to be related to the release mechanism from a polymer scaffold [47], the nano-sized network in the CHP scaffold might be a regulator, which determines the release rate from the scaffold. Among developing hydrogel scaffolds, the CHP nanogel-hydrogel shown in this study could be a novel hydrogel scaffold, whose physicochemical property was designed for the functional peptide to maintain the three-dimensional structure promoting the bioactivity of the peptide that helps for the peptide to work as a bone anabolic molecule.

## 4. Materials and Methods

### 4.1. Preparation Method for the Scaffold Materials

Two types of CHP nanogels were synthesized as previously described [27]. Both CHP nanogel materials were dissolved in super dehydrated dimethylsulfoxide (DMSO; FUJIFILM Wako Pure Chemical Corporation, Osaka, Japan), followed by the addition of 4-(4,6-dimethoxy-1,3,5-triazin-2-yl)-4-methyl-morpholinium chloride (DMT-MM; Tokyo Chemical Industry Co., Ltd., Tokyo, Japan). Then, N,N-Diisopropylethylamine and acrylic acid (DIPEA; Tokyo Chemical Industry Corp.) were added, and the mixture was stirred for 22 h. After stirring, the liquid was collected and dialyzed with MilliQ water, and DMSO for 4 days. The degree of substitution of the acryloyl groups in both CHP nanogel materials is as follows: The CHP-A and the CHP-OA nanogels (21 acryloyl groups per 100 monosaccharides) and PEG-SH were dissolved in 10× phosphate-buffered saline (PBS; FUJIFILM Wako Pure Chemical Corp.), then mixed and gelatinized to form the gels (Figure 1B). A cylindrical mold with a 6 mm diameter and 3 mm thickness was used to fabricate both CHP scaffolds. A 4 mm diameter mold was used, and solidified hydrogels were used to make the mold porous by the freeze–thaw method so that cells could easily penetrate inside, as previously described [27].

### 4.2. Experimental Animal

Twenty 5-week-old male BALB/c mice (CLEA, Tokyo, Japan) were used for experiment 1 (Expt. 1), and twenty-four 5-week-old male C57BL/6J mice (CLEA, Tokyo, Japan) were used for experiment 2 (Expt. 2). The mice were kept in the Central Breeding Room of the Experimental Animal Center, Tokyo Medical and Dental University where the experiments were conducted.

### 4.3. In Vivo Experimental Design

In Expt. 1 (Figure 7) and Expt. 2 (Figure 8), the quantity of the reagents was determined with reference to our previous study [27]. As shown below, there were a total of five experimental groups (*n* = 4) in Expt. 1: no scaffold group; CHP-A alone and CHP-OA alone groups; and CHP-A and CHP-OA separately impregnated with a signaling molecule, BMP-2.

(1)No scaffold (defects only);(2)CHP-A;(3)CHP-A + BMP-2 (2 µg; Bioventus LLC, Durham, NC, USA);(4)CHP-OA;(5)CHP-OA + BMP-2 (2 µg; Bioventus LLC, Durham, NC, USA).

Reagents for Expt. 1: BMP-2 was adjusted to 1 µg/µL. LF6 buffer (5 mM monosodium glutamate, 2.5% glycine, 0.5% sucrose, 0.01% polysorbate 80, pH 4.5) was used for dilution. Two µL of BMP-2 was dropped onto scaffolds as described above and allowed to fully impregnate overnight at 4 °C. Experimental mice were injected subcutaneously with medetomidine hydrochloride (0.5 mg/kg, Domitor^Ⓡ^; Zenoaq, Fukushima, Japan), followed 15 min later by ketamine hydrochloride (50 mg/kg, Ketalar^Ⓡ^; Sankyo, Tokyo, Japan), and then the mice were sedated [27]. After sedation, the frontal skin of the mice was incised to expose the calvaria, and a 4 mm-diameter bone defect was created using a Biopsy punch (Kai Medical, Gifu, Japan). Scaffolds were placed over the bone defect, the scalp was sutured, and atipamezole (0.75 mg/kg, Antisedan^Ⓡ^; Zenoaq, Fukushima, Japan), as an antagonist of medetomidine hydrochloride, was injected subcutaneously to shorten the sedation time for mice.

Exp. 2 was conducted by dividing each scaffold into three groups according to the different combinations of signaling molecules (BMP-2 or/and OP3-4), which were impregnated in CHP-A and CHP-OA scaffolds, for a total of six experimental groups (*n* = 4). Both scaffolds were separately used to impregnate with signaling molecules as described below:(1)CHP-A + BMP-2 (2 µg; Bioventus LLC, Durham, NC, USA);(2)CHP-A + BMP (2 µg) + Vehicle (solvents for BMP-2 and RANKL-binding peptide: the component ratio of the vehicle; NaOH (0.004 mol)/2.35 µL, DW/2.629 µL, NaOH (1 mol)/0.021 µL);(3)CHP-A + BMP-2 (2 µg) + OP3-4 (0.66 mg; Atlantic peptides, PA, USA, dissolved in the above RANKL-binding peptide solvent);(4)CHP-OA + BMP-2 (2 µg);(5)CHP-OA + BMP-2 (2 µg) + Vehicle;(6)CHP-OA + BMP-2 (2 µg) + OP3-4 (0.66 mg).

Reagent adjustment for Expt. 2: BMP-2 was adjusted to 1 µg/µL and LF6 buffer (5 mM monosodium glutamate, 2.5% glycine, 0.5% sucrose, 0.01% polysorbate 80, pH 4.5) was used for dilution. 2 µL of BMP-2, 5 µL of OP3-4 solvent or 5 µL of OP3-4 was dropped onto both CHP scaffolds. The methods of surgery were described above.

### 4.4. Radiographic Assessments

In vivo micro-computed tomography (μCT: R_mCT2; Rigaku, Tokyo, Japan) was performed at 90 kV, 160 μA, and field of view (ψ10 mm × H 10 mm) was used for analysis. Three-dimensional tomographic reconstructed images of the bone were acquired by the above-mentioned μCT system. The regenerated bone area was then measured using the Image J software (version 1.50 i; NIH, Bethesda, MD, USA) with the inner side of a 3.5 mm circle. At week 4 after scaffold placement, the mice were sedated with medetomidine hydrochloride (0.5 mg/kg, Domitor^Ⓡ^; Zenoaq, Fukushima, Japan), euthanized, and calvarias were removed and immersion-fixed overnight in 3.7% phosphate-buffered formalin solution (FUJIFILM Wako Pure Chemical Corporation, Osaka, Japan). The calvarias were then washed with PBS (FUJIFILM Wako Pure Chemical Corporation, Osaka, Japan) to remove formalin, and then soft X-ray imaging was performed using SOFTEX (SRO-M50, SOFRON, Kanagawa, Japan) at 27 kV, 4 mA. Cortical bone content and other bone parameters were measured with a peripheral quantitative computed tomography (pQCT; XCT Research SA+, Stratec Medizintechnik GmbH, Pforzheim, Germany). The region of interest (ROI) of 2.45 mm × 1.37 mm was set and measured, and total bone mineral content and bone mineral density were obtained.

### 4.5. Histological Assessments and Bone Histomorphometry

The calvarias were cut and embedded in the SCEM compound (Section-Lab Co., Ltd., Hiroshima, Japan) after euthanization. Thereafter, cranial samples were frozen into blocks at −100 °C in the freezer. The sections were stained using the von Kossa method for the calcified tissue. To measure bone formation activity, calcein (Sigma-Aldrich, St. Louis, MO, USA) was injected subcutaneously on day 10, and alizarin (Dojindo, Kumamoto, Japan) was injected on day 18 after surgery (Expt. 2). All fluorescent reagents were administered at a dose of 20 mg/kg (dose solution volume; 0.1 mL/10 g body weight) after surgery. Fluorescence images were taken using an inverted fluorescence microscope (FSX100; Olympus, Tokyo, Japan), and the total labeled bone area was calculated using ImageJ software with an ROI of 3.47 mm × 1.45 mm to measure bone formation activity.

### 4.6. Statistical Analysis

SPSS Version 27.0 for Windows (IBM Corp, Armonk, NY, USA) was used for the statistical analysis of quantitative data of bone measurements. Normality was assessed using the Shapiro–Wilk test. Comparisons between experimental groups were made by two-way ANOVA, and Tukey’s test was used for multiple comparisons. Data were expressed as the mean ± standard deviation. *p* values of <0.05 were considered to indicate a statistically significant difference.

## 5. Conclusions

In our study, we first confirmed that the CHP-A nanogel, which was expected to have a fast degradation rate from its chemical structure, degraded quicker than the CHP-OA nanogel. We next compared these two scaffolds, CHP-A and CHP-OA nanogel-hydrogels, both of which have a chaperone function. Our results clarified that both scaffolds showed good osteoconductive ability when used as a scaffold of both BMP-2 and the functional peptide OP3-4. The amount of regenerated bone in both CHP-A and -OA was similar when impregnated with the larger-sized BMP-2 molecule alone compared to the OP3-4 peptide; however, a higher amount of regenerated bone was obtained in the CHP-A group compared to the CHP-OA group only when combined with the smaller-sized molecule, OP3-4 peptide. These results suggest that the release rate of OP3-4 peptide was higher in the CHP-A group than in the CHP-OA group, while the release rate of BMP-2 was similar in both groups. The difference in regenerated bone mass might be explained by the molecular size-dependent release from the nanogel network. Further studies are necessary to clarify whether the affinity of a signaling molecule to the scaffold and the release profile from the scaffold would change depending on the molecular size of OP3-4 peptide and BMP-2.

## Figures and Tables

**Figure 1 ijms-24-09751-f001:**
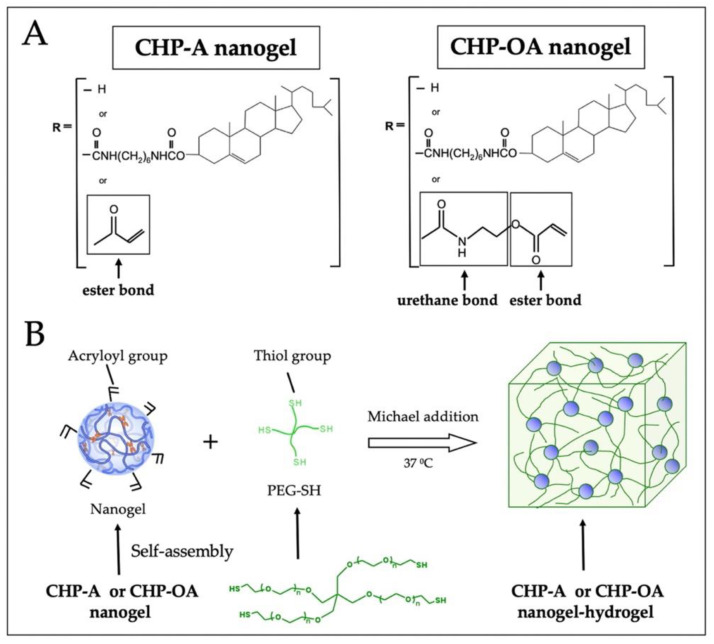
Schematic diagram of CHP-A and CHP-OA nanogel-hydrogels. (**A**) chemical structure of CHP-A and CHP-OA nanogel-hydrogels. (**B**) Schematic diagram of gelation process of CHP-A and CHP-OA nanogel-hydrogels. CHP, cholesterol-bearing pullulan; PEG-SH, 4-armed polyethylene glycol thiol.

**Figure 2 ijms-24-09751-f002:**
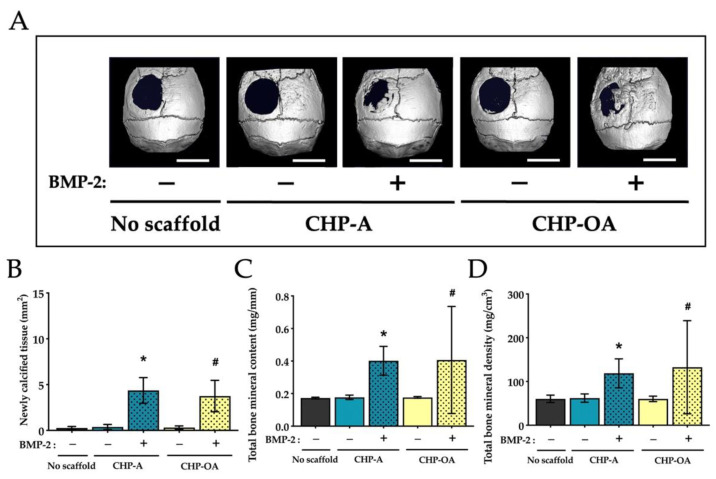
Micro-CT and pQCT analyses of scaffold material groups in a calvarial defect model. (**A**) Representative reconstruction images of μCT in the calvarial defect site at week 4 after scaffold placement. +/−: use with or without BMP-2. Scale bar = 4 mm. (**B**) Quantitative analysis of newly calcified tissue area in the calvarial defect site at week 4 after scaffold placement. (**C**,**D**) Quantitative analyses of bone measurements using pQCT in the calvarial defect site. Normality was analyzed by the Shapiro–Wilk test. The comparisons among experimental groups were performed using ANOVA and Tukey HSD test. Values are expressed as the mean ± SD. *: *p* < 0.05 vs. CHP-A alone group; #: *p* < 0.05 vs. CHP-OA alone group.

**Figure 3 ijms-24-09751-f003:**
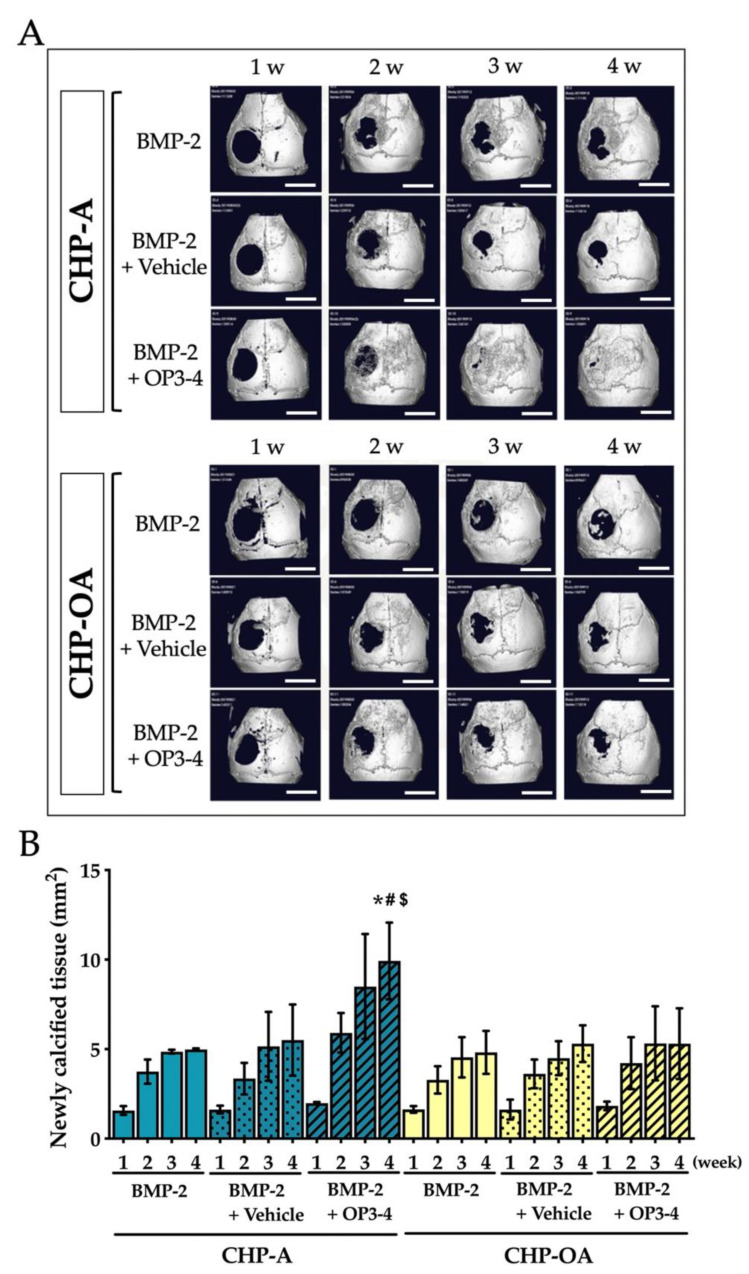
In vivo µCT analysis in a calvarial defect model. (**A**) Representative reconstruction μCT images of newly calcified tissue in the calvarial defect site at week 1, 2, 3, and 4 after scaffold placement. Scale bar = 4 mm. (**B**) Quantitative analysis of newly calcified tissue area in the calvarial defect site at week 1, 2, 3, and 4 after scaffold placement. Vehicle: solvents for BMP-2 and the RANKL-binding peptide. Green and yellow graphs represent the data when CHP-A and CHP-OA scaffolds were used, respectively. Normality was analyzed by the Shapiro–Wilk test. ANOVA and Tukey HSD test were used for the multiple comparisons. Values are expressed as the mean ± SD. *: *p* < 0.05 vs. BMP-2 + CHP-A group on week 4; #: *p* < 0.05 vs. BMP-2 + Vehicle + CHP-A group on week 4; and $: *p* < 0.05 vs. BMP-2 + OP3-4 + CHP-OA group on week 4.

**Figure 4 ijms-24-09751-f004:**
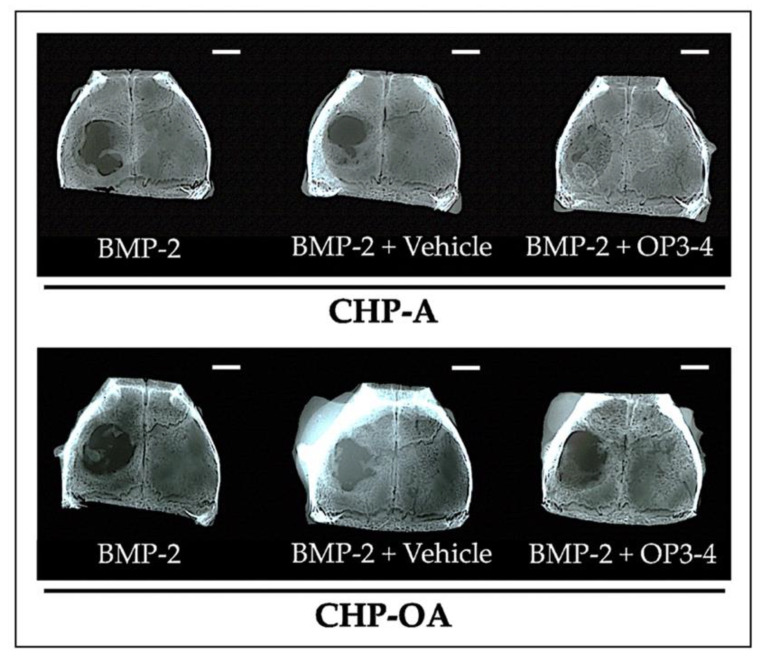
Soft X-ray images after scaffold placement in a calvarial defect model. Representative images of soft X-ray in the calvarial defect model. Vehicle: solvents for BMP-2 and the RANKL-binding peptide. Scale bar = 2 mm.

**Figure 5 ijms-24-09751-f005:**
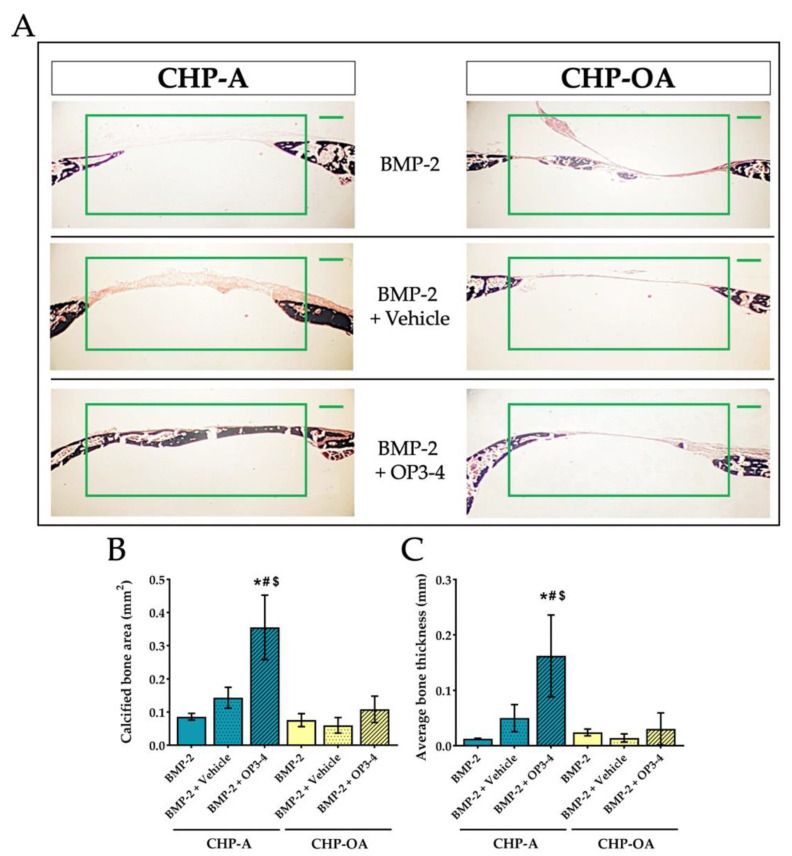
Changes of calcified tissue in a calvarial defect model. (**A**) Von Kossa-stained images of the calvarial bone. The green rectangle in the figure represents the region of interest (ROI) for measurements. Scale bar = 500 μm. (**B**,**C**) Quantitative analyses of calcified bone at the ROI of calvarial bone defect site. Vehicle: solvents for BMP-2 and the RANKL-binding peptide. Normality was analyzed by the Shapiro–Wilk test. The comparisons among experimental groups were performed using ANOVA and Tukey HSD test. Values are expressed as the mean ± SD. *: *p* < 0.05 vs. BMP-2 + CHP-A group; #: *p* < 0.05 vs. BMP-2 + Vehicle + CHP-A group; and $: *p* < 0.05 vs. BMP-2 + OP3-4 + CHP-OA group.

**Figure 6 ijms-24-09751-f006:**
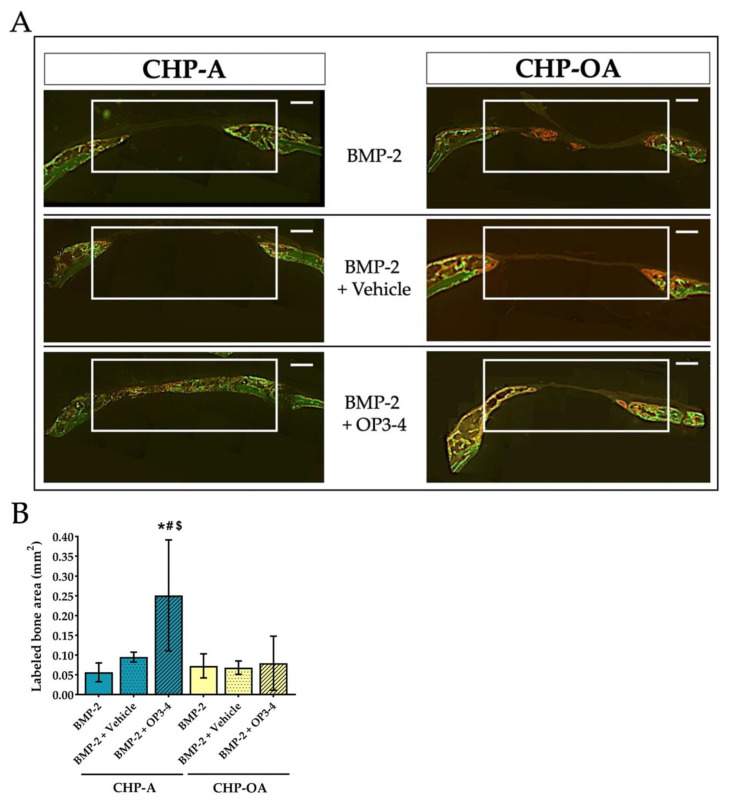
Bone formation analysis in a calvarial defect model. (**A**) Representative fluorescence images of undecalcified sections at the calvarial defect site. The white rectangle in the figure represents the ROI for measurement. Green calcein fluorescence was injected on day 10; red alizarin fluorescence was injected on day 18 before euthanization. Scale bar = 500 μm. (**B**) Quantitative analysis of bone histomorphometry at the ROI of the bone calvarial defect site. Labeled area: the sum of calcein and alizarin labeled area in the fluorescence images. Vehicle: solvents for BMP-2 and the RANKL-binding peptide. Normality was analyzed by the Shapiro–Wilk test. The comparisons among experimental groups were performed using ANOVA and Tukey HSD test. Values are expressed as the mean ± SD. *: *p* < 0.05 vs. BMP-2 + CHP-A group; #: *p* < 0.05 vs. BMP-2 + Vehicle + CHP-A group; and $: *p* < 0.05 vs. BMP-2 + OP3-4 + CHP-OA group.

**Figure 7 ijms-24-09751-f007:**
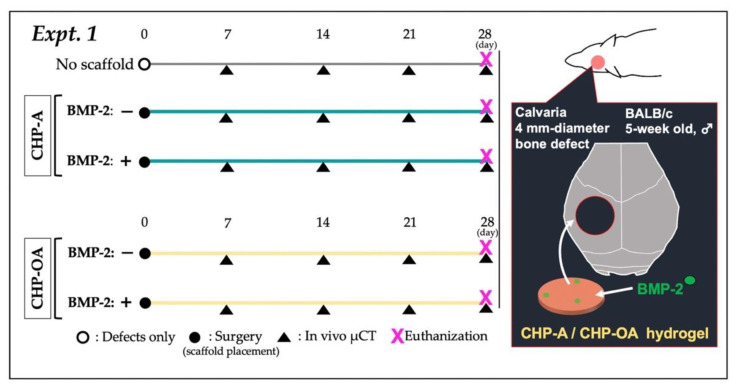
In vivo protocol of experiment 1. Black circle: surgery for placing scaffolds; black triangle: in vivo µCT imaging.

**Figure 8 ijms-24-09751-f008:**
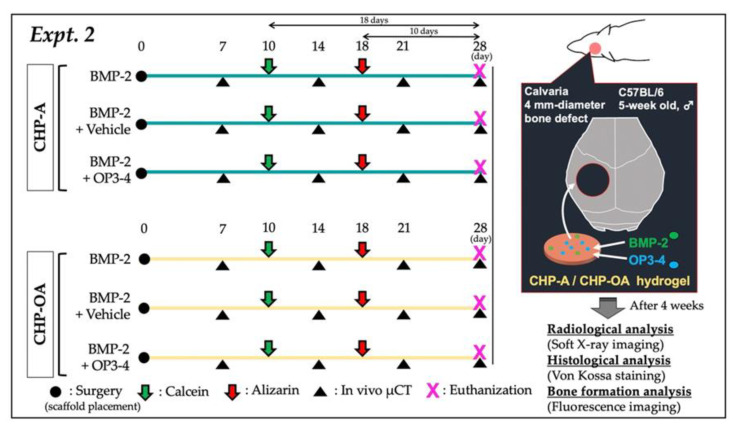
In vivo protocol of experiment 2. Black circle: surgery for placing scaffolds; black triangle: in vivo µCT imaging; green arrow: calcein injection; red arrow: alizarin injection.

## Data Availability

Not applicable.

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
