# Peer review of "Comparison of Osteoconductive Ability of Two Types of Cholesterol-Bearing Pullulan (CHP) Nanogel-Hydrogels Impregnated with BMP-2 and RANKL-Binding Peptide: Bone Histomorphometric Study in a Murine Calvarial Defect Model"

_ijms, 2023, doi:10.3390/ijms24119751_

Round 1

Reviewer 1 Report

Dear Editor,

In this manuscript entitled " Comparison of Osteoconductive Ability of Two Types of Cholesterol-Bearing Pullulan (CHP) Nanogel Scaffolds Impregnated with BMP-2 and RANKL-Binding Peptide: Bone Histomorphometric Study in a Murine Calvarial Defect Model" seems interesting, but there are some issues that need clarification or correction.

1- For each research method, it is necessary to expand the discussion. Please add schematic diagram for this study.

2- Please add some lines to indicate the novelty of your study, compare the results with that of the literature and emphasize the novelty of this study.

3-Line 265: please explain with more details.

4- In the conclusion, the performance findings of the research should have been summarized the innovations and future scope of the work should be highlighted more.

Author Response

Thank you very much for your positive comment. 

Please find out the attached file, which shows our response to your comments.

Reviewer 2 Report

Xie et al have presented the results on the Comparison of Osteoconductive Ability of Two Types of Cho- 3 lesterol-Bearing Pullulan (CHP) Nanogel Scaffolds.

Below are my comments for the improvement of the manuscript.

1. The intro part is very short, it must be of sufficient length covering new trends, research question, background and novelity

2. The result and discussion part needs significant improvement as the results need explanation in line with the recent studies

3. The conclusion is also too short. I suggest to re-write it and elaborate the main results in it.

English needs moderate corrections

Author Response

Thank you very much for your positive comment.

Please find out the attached file, which shows response to your comments.

Reviewer 3 Report

The article of " Comparison of Osteoconductive Ability of Two Types of Cholesterol-Bearing Pullulan (CHP) Nanogel Scaffolds Impregnated with BMP-2 and RANKL-Binding Peptide: Bone Histomorphometric Study in a Murine Calvarial Defect Model” is a very interesting topic. Authors had published a lot of papers about CHP with BMP-2 and RANKL-Binding Peptide in bone regeneration application. The manuscript is well written and easy to understand.

 Several areas of the article need to be improved.

1.      Why not choose the same mice strain in experiment 1 and experiment 2 of animal studies?

2.      Line 67 “An osteoclastogenesis assay was performed in vitro and revealed a slower release of RANKL-binding peptide from CHP-OA nanogel hydrogel, compared to CHP-A nanogel hydrogel [26]”

But reference 26 did not claim a slower release of RANKL-binding peptide from CHP-OA nanogel hydrogel, compared to CHP-A nanogel hydrogel.

3.      What is the release prolife of BMP-2 and OP3-4 from CHP-OA and CHP-A?

Author Response

(The authors gave the same response as above.)

Round 2

Reviewer 1 Report

Accept in present form

 Moderate editing of English language required

Reviewer 2 Report

The authors have shown a significant improvement in the revised version. The manuscript in its present form can be published online.

Before the manuscript gets online, I would suggest to check for minor English editing errors.

Reviewer 3 Report

The authors have responded in detail to each of the comments of the reviewers.  It is suitable to publish now.

Minor editing of English language required.